# Nanopore Deep Sequencing as a Tool to Characterize and Quantify Aberrant Splicing Caused by Variants in Inherited Retinal Dystrophy Genes

**DOI:** 10.3390/ijms25179569

**Published:** 2024-09-03

**Authors:** Jordi Maggi, Silke Feil, Jiradet Gloggnitzer, Kevin Maggi, Ruxandra Bachmann-Gagescu, Christina Gerth-Kahlert, Samuel Koller, Wolfgang Berger

**Affiliations:** 1Institute of Medical Molecular Genetics, University of Zurich, 8952 Schlieren, Switzerland; maggi@medmolgen.uzh.ch (J.M.); feil@medmolgen.uzh.ch (S.F.); gloggnitzer@medmolgen.uzh.ch (J.G.); kmaggi@medmolgen.uzh.ch (K.M.); koller@medmolgen.uzh.ch (S.K.); 2Institute of Medical Genetics, University of Zurich, 8952 Schlieren, Switzerland; ruxandra.bachmann@mls.uzh.ch; 3Department of Molecular Life Sciences, University of Zurich, 8057 Zurich, Switzerland; 4Neuroscience Center Zurich (ZNZ), University and ETH Zurich, 8057 Zurich, Switzerland; 5Department of Ophthalmology, University Hospital Zurich and University of Zurich, 8091 Zurich, Switzerland; christina.gerth-kahlert@usz.ch; 6Zurich Center for Integrative Human Physiology (ZIHP), University of Zurich, 8057 Zurich, Switzerland

**Keywords:** splicing, splice variant, Nanopore, pseudoexon, minigene assay, exon skipping, inherited retinal dystrophies (IRDs), long-read sequencing

## Abstract

The contribution of splicing variants to molecular diagnostics of inherited diseases is reported to be less than 10%. This figure is likely an underestimation due to several factors including difficulty in predicting the effect of such variants, the need for functional assays, and the inability to detect them (depending on their locations and the sequencing technology used). The aim of this study was to assess the utility of Nanopore sequencing in characterizing and quantifying aberrant splicing events. For this purpose, we selected 19 candidate splicing variants that were identified in patients affected by inherited retinal dystrophies. Several in silico tools were deployed to predict the nature and estimate the magnitude of variant-induced aberrant splicing events. Minigene assay or whole blood-derived cDNA was used to functionally characterize the variants. PCR amplification of minigene-specific cDNA or the target gene in blood cDNA, combined with Nanopore sequencing, was used to identify the resulting transcripts. Thirteen out of nineteen variants caused aberrant splicing events, including cryptic splice site activation, exon skipping, pseudoexon inclusion, or a combination of these. Nanopore sequencing allowed for the identification of full-length transcripts and their precise quantification, which were often in accord with in silico predictions. The method detected reliably low-abundant transcripts, which would not be detected by conventional strategies, such as RT-PCR followed by Sanger sequencing.

## 1. Introduction

Inherited retinal dystrophies (IRDs) constitute a group of conditions affecting the retina, which is a thin layer of neuronal cells lining the back of the eye. These disorders are characterized by deterioration or congenital dysfunction of the photoreceptors or the retinal pigment epithelium, which ultimately results in impaired vision or blindness. The genetic basis of IRDs is highly heterogeneous, with variants in over 300 loci known to lead to these disorders (RetNet, https://sph.uth.edu/RetNet/ accessed on 28 June 2024). Molecular diagnostics for this group of conditions is further aggravated by high clinical (phenotypic) heterogeneity [1]. Despite this complexity, recent studies (2018–2022) reported an overall diagnostic yield for mixed IRDs of 64.2% [2]. This figure improves to 73.5% when considering exclusively studies using exome sequencing (WES) [2].

The Human Gene Mutation Database (HGMD) Professional (https://my.qiagendigitalinsights.com/bbp/view/hgmd/pro/start.php, accessed on 1 July 2024) contains 519,879 unique variant entries in 17,609 human loci. Coding variants are the most common type included in HGMD (82.5%), followed by variants affecting splicing (8.3%), and structural variants (7.7%) [3]. In IRDs, the contribution of splicing variants to disease has been reported to be similar [4,5]. Weisschuh et al. recently reported that WGS allowed for the identification of a molecular diagnosis in 74.1% of their 1000 IRD and inherited optic neuropathies patients cohort [5]. Among the molecularly diagnosed patients, 1097 unique pathogenic variants were identified; of these, 81.0% were coding variants (548 missense variants, 139 nonsense variants, 173 frameshift variants, 27 in-frame insertions or deletions, 1 start loss variant, and 1 stop loss variant), 11.6% splicing variants (70 canonical splice site variants, 44 non-canonical splice site variants, and 13 deep-intronic splice variants), 0.5% regulatory variants, and 6.5% structural variants [5].

Whole-gene sequencing and functional characterization of candidate splicing variants in *ABCA4* have been the subject of extensive efforts for the resolution of missing heritability in IRDs, which led to the identification of many pathogenic variants that WES would not detect [6,7,8,9,10,11,12,13,14,15,16,17]. These studies have identified and characterized many deep-intronic and non-canonical splice site variants affecting splicing. Similarly, a recent study curated pathogenicity classification for the 2246 *ABCA4* variants reported within the LOVD database [18]. They classified 1248 variants to be likely pathogenic or pathogenic; among these, 254 (20.4%) variants may affect splicing, including 52 (4.2%) non-canonical or deep-intronic variants [18]. All of these studies demonstrated that splicing variants remain underreported in the literature due to the difficulty in predicting their effects in silico and the inability to detect them, as deep-intronic regions are not covered by WES.

A multitude of in silico tools have been developed to predict the impact of a variant on splicing [19,20,21,22,23,24,25,26]. The performance of some of these tools has been benchmarked with splicing variants in the *ABCA4*, *MYBPC3*, and *NF1* genes that had been functionally characterized [27,28]. These studies found that deep learning tools (areas under the curve (AUCs) of 0.72–0.99) often outperform classical machine learning (AUCs of 0.69–0.80) and motif-based tools (AUCs of 0.72–0.86) [27,28]. However, Riepe et al. found that the variant context played an important role in determining which was the best-performing tool [28].

Splicing prediction tool scores can help in genetic testing during variant filtering and prioritization. However, in the absence of functional assays, these scores are insufficient for categorizing variants outside of canonical splice sites as likely pathogenic or pathogenic [29]. Depending on the accessibility of the tissue expressing the mutated gene, reverse-transcriptase (RT)-PCR or RNA-seq can provide insights into potential splicing defects in patient-derived cells [30]. Alternatively, minigene constructs containing the genomic region surrounding the candidate variant can enable the identification and characterization of aberrant splicing [31]. Many genes associated with IRD pathogenesis are not stably expressed in readily accessible tissues, such as whole blood [5], which makes minigene assays particularly interesting for variants in these genes [6,17,32,33,34,35,36,37,38,39].

The aim of this study was to test the performance of Nanopore deep sequencing for the characterization and quantification of variant-induced aberrant splicing events. For this purpose, we selected 19 candidate variants that may affect splicing, which were detected in patients affected by IRDs. We functionally characterized these variants using minigene assay or patient-derived peripheral blood RT-PCR, combined with Nanopore sequencing, to identify and quantify alternative splicing products. The results of these functional readouts were compared to the predictions obtained with in silico tools. Thirteen of the nineteen variants led to aberrant splicing events and may have a negative impact on protein function.

## 2. Results

### 2.1. Variant Selection

Nineteen rare variants that may affect splicing were selected for functional characterization by Nanopore sequencing (Table 1). These variants occur in genes expressed in the retina, previously associated with an IRD phenotype. The candidate variants were identified in IRD patients; some of these patients remained undiagnosed after WES, whole-gene long-range PCR sequencing (LR-PCR), WGS, or a combination of these methods, as reported in previous studies [40,41]. Demographic data of these patients is summarized in Appendix A.

To our knowledge, only two of these variants (NM_172240.2:c.1033-327T>A and NM_001034853.1:c.1415-9A>G) have been functionally tested previously for their effect on splicing [39,42]. As a positive control for an aberrant splicing event, we included the non-canonical splice site variant in *RPGR,* that we previously described, to lead to an out-of-frame insertion of 8 nucleotides of intron 11 as a positive control [39]. The selected variants include 3 missense (*CHM*:c.1413G>C (p.(Gln471His)), *FZD4*:c.313A>G (p.(Met105Val)), and *REEP6*:c.517G>A (p.(Val173Ile))), 3 synonymous (*ABCA4*:c.573C>T, *ABCA4*:c.5586T>A, and *PROM1*:c.2358C>T), 4 canonical splice site (*KIF11*:c.1875+2T>A, *PDE6C*:c.864+1G>A, *POC1B*:c.677-2A>G, and *PROM1*:c.2490-2A>G), 5 non-canonical splice site (*ATF6*:c.1096-15G>A, *ATF6*:c.1534-9A>G, *CACNA1F*:c.2239+5C>G, *IMPG2*:c.3423-7_3423-4del, and *RPGR*:c.1415-9A>G), and 4 deep-intronic (*OCA2*:c.574-53C>G, *POC1B*:c.1033-327T>A, *TIMP3*:c.205-3117T>C, and *USH2A*:c.652-22287T>C) sequence alterations.

**Table 1 ijms-25-09569-t001:** Nineteen candidate splicing variants in 15 IRD-associated genes. Classification according to American College of Medical Genetics and Genomics (ACMG) guidelines from Varsome and Franklin.

Gene	Variant (cNomen)	gnomAD All (%)	ACMG	LOVD	ClinVar	HGMD	Ref.	Clinical Phenotype
*ABCA4*	NM_000350.2:c.573C>T	0.004	LB/VUS	-	LB	-	-	MD/OCA
*ABCA4*	NM_000350.2:c.5586T>A	0.011	LB/LB	VUS	LB	-	-	RD
*ATF6*	NM_007348.3:c.1096-15G>A	NA	LB/VUS	-	-	-	-	ACHR
*ATF6*	NM_007348.3:c.1534-9A>G	0.002	LP/VUS	-	-	-	-	ACHR
*CACNA1F*	NM_005183.4:c.2239+5C>G	NA	LB/VUS	-	-	-	-	RD
*CHM*	NM_000390.4:c.1413G>C	NA	LP/VUS	-	-	DM	[43]	RP/CHM
*FZD4*	NM_012193.4:c.313A>G	0.002	P/P	P	P	-	-	EVR
*IMPG2*	NM_016247.4:c.3423-7_3423-4del	0.010	VUS/LP	P	VUS/LP/P	DM	[44]	MD
*KIF11*	NM_004523.3:c.1875+2T>A	NA	LP/LP	-	-	-	-	EVR
*OCA2*	NM_000275.3:c.574-53C>G	0.500	B/LB	-	-	-	-	MD/OCA
*PDE6C*	NM_006204.3:c.864+1G>A	NA	P/P	P	LP	DM	[45]	COD
*POC1B*	NM_172240.2:c.677-2A>G	0.001	LP/LP	-	-	DM?	[46]	CRD
*POC1B*	NM_172240.2:c.1033-327T>A	0.005	VUS/VUS	P	-	DM	[42]	CRD
*PROM1*	NM_006017.3:c.2358C>T	0.045	B/B	B	B	-	-	MD
*PROM1*	NM_006017.3:c.2490-2A>G	0.012	P/P	P	VUS/LP/P	DM?	[5,46]	STGD
*REEP6*	NM_001329556.3:c.517G>A	NA	VUS/VUS	-	-	-	-	RP
*RPGR*	NM_001034853.1:c.1415-9A>G	NA	VUS/LP	-	LP	DM	[39]	RP
*TIMP3*	NM_000362.4:c.205-3117T>C	NA	LB/VUS	-	-	-	-	VMD
*USH2A*	NM_206933.2:c.652-22287T>C	0.083	LB/VUS	-	-	-	-	RP

Abbreviations: cNomen, Human Genome Variation Society (HGVS) cDNA-level nucleotide change nomenclature; gnomAD all (%), genome aggregation database v2.1.1 overall minor allele frequency in percentage; LOVD, Leiden Open Variation Database; ClinVar, Clinical Variation database; HGMD, Human Gene Mutation Database; Ref., reference; VUS, variant of unknown significance; P, pathogenic; LP, likely pathogenic; LB, likely benign; B, benign; DM, disease-causing mutation; DM?, disease-causing mutation?; and NA, not available.

### 2.2. Splicing Predictions for Candidate Variants

All variants selected for functional characterization were assessed for possible effects on splicing by in silico predictions using several algorithms, including those present in Alamut^®^ Visual Plus, SpliceAI [19], and Pangolin [20] (Appendix A). Table 2 lists the findings, and the most likely variant-caused aberrant splicing events based on these predictions. The average variant-induced difference in splice site strengths computed by tools in Alamut Visual Plus will be referred to as the “Effect score”. Appendix A provides a visual representation of the splicing predictions from Alamut^®^ Visual Plus for each variant and its flanking sequences.

Based on the predictions, the variants in *ABCA4*, *CACNA1F*, *CHM*, *OCA2*, *PDE6C*, and *PROM1* are expected to lead to partial exon skipping, as they only affect natural acceptor and/or donor sites. Similarly, the *ATF6*, *IMPG2*, *KIF11*, *POC1B* (NM_172240.2:c.677-2A>G), and *REEP6* variants affect the natural splice sites; however, they may also influence cryptic splice sites, by either creating new or strengthening pre-existing sites, and most of them impact the exonic splicing enhancer (ESE) to exonic splicing silencer (ESS) binding sites ratios. As a result, these variants are predicted to cause partial exon skipping and/or partial usage of an alternative (cryptic) acceptor or donor splice site. In silico predictions suggest that the *FZD4*, *POC1B* (NM_172240.2:c.1033-327T>A), and *RPGR* variants create cryptic acceptor sites and could lead to partial usage of these alternative cryptic acceptor sites. Variant NM_172240.2:c.1033-327T>A concomitantly abolishes a natural acceptor splice site of the noncoding *POC1B* transcript NR_037659.2. Finally, the *TIMP3* and *USH2A* variants only slightly affect nearby cryptic splice sites; however, both decrease the ESE/ESS ratio of the predicted pseudoexon, making its inclusion in the transcript more likely when compared to the reference sequence.

### 2.3. Minigene Assays for Candidate Variants

At least one minigene plasmid was successfully constructed for each variant (Table 3), except for the *KIF11* variant. The *KIF11* variant was functionally tested using whole blood cDNA from the affected family (see Section 2.4). Most minigene constructs (16/19) were based on the pcDNA3_RHO_ex3-5_plasmid (refer to Materials and Methods Section 4.5). The *FZD4* minigene contains the entire *FZD4* locus. For the *ATF6* variants, minigenes containing exons 1, 2, and 9 or 13, with flanking introns, were constructed from three PCR products. For the *IMPG2* variant, a large (including exons 15–18) and a minimal (including only exon 17) minigene were created. A circular view of the features of each plasmid is available in Appendix A.

The expected major (WT) transcript (highlighted in green under the coverage plots in Figure A1, Figure A2, Figure A3, Figure A4, Figure A5, Figure A6, Figure A7, Figure A8, Figure A9, Figure A10, Figure A11, Figure A12, Figure A13, Figure A14, Figure A15, Figure A16, Figure A17, Figure A18, Figure A19 and Figure A20) was identified in all assays for the reference minigene. Its relative abundance as measured by Nanopore sequencing, however, varied greatly from only 1.1% to 98.7% of total reads. We also identified the transcript composed only of *RHO* exons in most reference minigene assays (10/16) at levels ranging from 1.6% to 80.9% (Figure A1, Figure A2, Figure A3, Figure A4, Figure A5, Figure A6, Figure A7, Figure A8, Figure A9, Figure A10, Figure A11, Figure A12, Figure A13, Figure A14, Figure A15, Figure A16, Figure A17, Figure A18, Figure A19 and Figure A20 and Table A1, Table A2, Table A3, Table A4, Table A5, Table A6, Table A7, Table A8, Table A9, Table A10, Table A11, Table A12, Table A13, Table A14, Table A15, Table A16, Table A17, Table A18, Table A19 and Table A20).

Reference minigenes that resulted in low abundance (<50%) of WT transcript included RHO_minigene_ATF6_int8-9 (32.5%, Figure A3 and Table A3), RHO_minigene_CACNA1F_int14-18 (11.2%, Figure A6 and Table A6), RHO_minigene_IMPG2_int15-18 (1.1%, Figure A9 and Table A9), RHO_minigene_OCA2_int5-7 (17.6%, Figure A11 and Table A11), and RHO_minigene_PROM1_int23-26 (3.7%, Figure A16 and Table A16). In all these cases, another transcript resulting from at least one exon-skipping event was present in the reference minigene splicing assay. These findings can be partially explained by the splice site strengths of the skipped exons (Appendix A). In fact, *ATF6* exon 9 has a relatively weak acceptor site (56%), *CACNA1F* exons 16 and 17 and *OCA2* exons 6 and 7 are flanked by very weak splice sites, and the donor splice site defining *IMPG2* exon 17 is relatively weak (44%). On the other hand, *RHO* exons 3 and 5 are characterized by strong donor and acceptor (average transformed Alamut scores of 75% and 67%, respectively). Exon skipping due to an imbalance in splice site strengths has been postulated previously in a similar study [17]. This may explain the presence of the transcript composed only by *RHO* exons in most minigene assays. Additionally, shorter transcripts are preferentially amplified during PCR amplification, which can lead to a bias.

The minigene assays revealed aberrant splicing events for 12/18 variants (Table 4). Six additional variants in *ABCA4* (NM_000350.2:c.573C>T and NM_000350.2:c.5586T>A), *FZD4* (NM_012193.4:c.313A>G), *OCA2* (NM_000275.3:c.574-53C>G), *TIMP3* (NM_000362.4:c.205-3117T>C), and *USH2A* (NM_206933.2:c.652-22287T>C) revealed no evidence of aberrant splicing (Figure A1, Figure A2, Figure A8, Figure A11, Figure A19 and Figure A20 and Table A1, Table A2, Table A8, Table A11, Table A19 and Table A20, respectively).

Generally, the relative abundance of WT transcript in variant minigene assays was reduced when aberrant splicing events were present. However, comparing the relative abundance of WT transcripts can be misleading. The differences in WT transcript abundance between reference and variant minigenes for the *CACNA1F* (Figure A6 and Table A6) and one of the *PROM1* (NM_006017.3:c.2490-2A>G, Figure A16 and Table A16) variants are small (−11.2% and −3.7%, respectively), but it is important to notice that the variant minigene showed complete depletion of the WT transcript. The delta value is low, only due to the low abundance of the WT transcript in the reference minigene assays in these cases. An overview of the gel electrophoresis results for RT-PCR products of each minigene can be accessed in Appendix A.

#### 2.3.1. Alternative Splice Sites (ATF6, PDE6C, POC1B, and RPGR)

The *ATF6* (NM_007348.3:c.1096-15G>A and NM_007348.3:c.1534-9A>G), *PDE6C* (NM_006204.3:c.864+1G>A), *POC1B* (NM_172240.2:c.677-2A>G), and *RPGR* (NM_001034853.1:c.1415-9A>G) variants were predicted to create or strengthen a cryptic acceptor or donor site and to weaken or disrupt the natural acceptor or donor splice site (Table 2). The minigene assays for these variants supported these predictions.

Variant NM_007348.3:c.1096-15G>A (*ATF6*) was functionally tested using two different minigene constructs: one based on the *RHO* backbone, and one containing the endogenous *ATF6* exons 1 and 2 inserted upstream of *ATF6* exon 9 (Table 3). The RHO_minigene_ATF6_int8-9 assay resulted in the identification of the predicted alternative acceptor splice site at position c.1096-13, being used in 2.8% of the variant minigene reads (Figure A3 and Table A3). Similarly, the ATF6_minigene_ex1-2-9 found the cryptic acceptor site in 6.1% of the reads. Additionally, this minigene assay identified another cryptic acceptor site at position c.159+275 (part of *ATF6* intron 2) in 3.0% of the reads (Figure A4 and Table A4). The reduction of WT transcript for both assays is relatively low (+1.4% and −25.2%, respectively), which suggests that this variant may have a mild effect on splicing. A mild effect could be expected based on in silico predictions, which anticipated a 7.1% Effect score and 17% SpliceAI reduction in natural splice site strength (Appendix A), and the creation of the c.1096-13 acceptor splice site with a 25% Effect score and 20% SpliceAI and Pangolin scores (Appendix A).

The ATF6_minigene_ex1-2-13 assay for variant NM_007348.3:c.1534-9A>G (*ATF6*) confirmed the activation of an alternative acceptor splice site at position c.1534-8 that was used in 77.3% of the reads (Figure A5 and Table A5). The relative abundance of the WT transcript was reduced from 87.5% in the reference minigene to 6.9% in the variant minigene. The c.1534-8 acceptor splice site was expected, based on in silico predictions, with a strength of 64.5% Effect score, 89% SpliceAI, and 82% Pangolin scores (Appendix A).

The *PDE6C* variant (NM_006204.3:c.864+1G>A) abolished the weak natural donor site and was expected to cause exon 4 skipping. The assay, however, identified the usage of two cryptic donor sites instead in the variant minigene (Figure A12 and Table A12); one was used in 65.5% of the reads and it corresponds to position c.864+128 in intron 4 (extending exon 4 by 128 nucleotides), and the second one is located in exon 4 at position c.801 (shortening exon 4 by 63 nucleotides), present in 28.6% of the reads.

The *POC1B* variant NM_172240.2:c.677-2A>G abolished the natural acceptor splice site of exon 7 and is predicted to create an alternative weak acceptor splice site (8.9% Effect score) at position c.684. Nanopore sequencing of the minigene cDNA confirmed the use of the predicted alternative acceptor splice site in 86.2% of the variant minigene reads, with no WT transcript detected (Figure A13 and Table A13).

The findings of the RHO_minigene_RPGR_int10-13 construct, characterized using gel electrophoresis and Sanger sequencing, were published in Koller et al., 2023 [39]. Briefly, we reported that the *RPGR* variant (NM_001034853.1:c.1415-9A>G) led to the extension of exon 12 by 8 nucleotides at the 5′ end in the vast majority of the transcripts. We also reported that partial exon 12 skipping was detected in the reference and variant minigene results. Nanopore sequencing of the minigene cDNAs revealed a similar, but more complex, set of transcripts (Figure A18 and Table A18). The alternative acceptor site is part of 70.5% of the transcripts in the variant minigene (transcripts T9, T10, and T11). Exon 12 skipping was found in 4.6% and 14.6% of reference and variant minigene transcripts (transcripts T4, T6, T12, and T13), respectively. A shorter exon 12 (using a cryptic acceptor site at position c.1427) was detected in 3.0% of the reference minigene transcripts (transcripts T5 and T8). Additionally, exon 11 was found to be alternatively spliced by using two alternative acceptor splice sites at positions c.1337 and c.1390. Finally, Nanopore sequencing revealed evidence of exon 13 skipping in the variant minigene transcripts T12 and T14, representing 1.4% of the reads.

#### 2.3.2. Exon Skipping in CHM and PROM1 Minigene Constructs

Exon skipping was found to be the main variant-induced aberrant splicing event by the *CHM* (NM_000390.4:c.1413G>C) and both *PROM1* (NM_006017.3:c.2358C>T and NM_006017.3:c.2490-2A>G) variants (Table 4).

The NM_000390.4:c.1413G>C (*CHM*) missense variant is located at the exon-intron boundary of *CHM* exon 11 and is predicted to weaken the natural donor splice site (Table 2). The sequencing results highlighted complete exon 11 skipping for the variant minigene (Figure A7 and Table A7).

The synonymous variant in exon 23 of *PROM1* (NM_006017.3:c.2358C>T) influences the exonic splicing enhancer (ESE) and silencer binding sequences (ESS) ratio, leading to the variant exon being more likely skipped during splicing (Table 2 and Appendix A). While the Effect score on natural splice sites is marginal, SpliceAI and Pangolin estimate exon splicing loss with a chance of about 27.7% (Appendix A). The assay revealed exon 23 was skipped in 40.9% of the variant minigene transcripts (Figure A15 and Table A15).

The assay using RHO_minigene_PROM1_int23-26 for variant NM_006017.3:c.2490-2A>G showed unexpectedly low levels of WT transcript (3.7% in reference and 0% in variant minigenes, Figure A16 and Table A16). However, *PROM1* exons 25 and 26 (NM_006017.3) are skipped in a subset of isoforms (NM_001145852.2, NM_001371408.1, NM_001371407.1, NM_001145850.2, NM_001145851.2, and NM_001145849.2). Therefore, the most abundant transcript in the reference minigene assay for variant NM_006017.3:c.2490-2A>G is part of the “normal” splicing of *PROM1* and could correspond to isoforms NM_001371407.1, NM_001145850.2, NM_001371408.1, or NM_001145852.2. The variant increased the likelihood of exon 25 skipping (12.0% in the variant minigene and 5.3% in the reference minigene, transcript T3, Figure A16 and Table A16).

#### 2.3.3. Multiple Aberrant Splicing Events in CACNA1F, IMPG2, and REEP6 Minigene Constructs

Assays for the *CACNA1F* (NM_005183.4:c.2239+5C>G), *IMPG2* (NM_016247.4:c.3423-7_3423-4del), and *REEP6* (NM_001329556.3:c.517G>A) variants revealed multiple aberrant splicing events, including exon skipping and cryptic splice site use (Table 4).

Based on in silico predictions, the NM_005183.4:c.2239+5C>G variant upstream of *CACNA1F* exon 16 is expected to cause partial exon skipping (Table 2) as SpliceAI computes a donor loss likelihood of 23% and Pangolin predicts splice loss with a score of 47% (Appendix A). Transcripts analysis revealed 18 unique transcripts represented by at least 0.5% of the reads (Figure A6 and Table A6). The main variant-induced aberrant splicing events were increased exon 16 skipping and alternative acceptor splice site for exon 17 (transcripts T1, T2, and T8) with 45.9% against 71.6% of reads in the reference versus variant minigene, respectively (Figure A6 and Table A6). The WT transcript could not be detected in the variant minigene.

Variant NM_016247.4:c.3423-7_3423-4del was functionally tested using two minigene constructs based on the *RHO* backbone: one containing *IMPG2* exons 16–18 and a minimal minigene containing only exon 17 and flanking introns (Table 3). The variant is predicted to weaken the natural acceptor site of exon 17 (−19.7% Effect score) and to strengthen a cryptic acceptor splice site (+1.1% Effect score) of 80 nucleotides upstream of exon 17 (Table 2 and Appendix A). The RHO_minigene_IMPG2_int15-18 minigene resulted in increased exons 16–17 skipping for the variant minigene (52.2% against 36.7% in the reference minigene; Figure A9 and Table A9). Sequencing of the cDNA from the reference minigene also highlighted unexpectedly low levels of WT transcript (1.1%). The results for the minimal minigene (RHO_minigene_IMPG2_int16-17) supported increased exon 17 skipping as a variant-dependent effect (67.6% against 42.2% in the reference minigene); however, they also revealed a transcript using the cryptic acceptor site located at c.3423-80 in 18.6% of the reads (Figure A10 and Table A10). The WT transcript was drastically reduced from 56.1% in the reference minigene, as opposed to 9.2% in the variant minigene (Figure A10 and Table A10).

The NM_001329556.3:c.517G>A missense variant lies at the exon-intron 4 boundary of *REEP6*. It severely weakens the natural donor splice site (−52.1% Effect score) and moderately strengthens a cryptic donor splice site, located at c.517+4 (+1.2% Effect score; Appendix A). The assay demonstrated evidence of exon 4 skipping (56.3% of the variant minigene reads), and the use of an alternative donor splice site located at c.517+43 in a transcript, representing 10.4% of the reads (Figure A17 and Table A17). The alternative donor splice site c.517+43 was predicted by SpliceAI and Pangolin with scores of 46% and 20%, respectively (Appendix A). The WT transcript was severely reduced in the variant minigene (10.6% against 66.6% in the reference minigene).

#### 2.3.4. Pseudoexon Inclusion in POC1B

Variant NM_172240.2:c.1033-327T>A is predicted to completely disrupt the natural acceptor site of exon 9 of the noncoding *POC1B* transcript NR_037659.2 and to create a stronger cryptic acceptor site, located 5 nucleotides upstream of it (c.1033-325). The minigene assay revealed that the WT transcript was strongly reduced in the variant minigene (43.3% against 91.3% in the reference minigene; Figure A14 and Table A14) and that the predicted novel acceptor site (c.1033-325) was used in 38.3% of the reads in combination with the pre-existing natural donor splice site from exon 9 of *POC1B* transcript NR_037659.2 (Figure A14 and Table A14). Therefore, a large portion of the transcripts include a pseudoexon (or an elongated version of exon 9 of NR_037659.2), which is not part of the “normal” splicing of the protein-coding transcripts. This variant has been functionally characterized previously with a minigene assay and patient-derived blood cDNA. Both assays revealed pseudoexon (or the elongated exon 9 of NR_037659.2) inclusion as a consequence of the variant [42].

### 2.4. Splicing Assays on Blood cDNA for KIF11 and CACNA1F

The *KIF11* variant (NM_004523.3:c.1875+2T>A) was functionally characterized using cDNA derived from the blood of three family members; the index patient and both parents. The index patient and the mother carry the variant heterozygously; the father is homozygous for the major allele at this position. The variant is located at the natural dinucleotide donor splice site of *KIF11* exon 14 and is predicted to lead to a 91% chance of donor loss and 6% chance of donor gain at position c.1785 by SpliceAI (Appendix A).

As expected, the sequencing results for the father revealed the WT transcript in at least 90.0% of reads (Figure 1 and Table 5). An additional 6.8% of the reads were lacking either exon 13 or 16 (transcripts T2–T5), which probably represents incomplete reads, as the PCR used primers binding to these exons for the amplification from cDNA. Therefore, it is likely that these reads represent incompletely sequenced transcripts.

Transcript quantification of the cDNA from the mother resulted in 41.5% of WT transcript (up to 63.1% if T2–T5 are considered incomplete WT transcripts), which was expected as the index and the mother are heterozygous for the variant. Sequencing allowed for the identification of the predicted alternative acceptor splice site at position c.1785 being used in 15.5% (transcript T6), and a transcript characterized by exon 14 skipping in 5.3% of the reads (transcript T7). Additionally, sequencing identified alternative donor or acceptor splice sites used for exons 15 and 16 (transcript T8). Finally, a minority of the reads (1.9%, transcript T9) were distinguished by exon 14 skipping and the inclusion of a pseudoexon from intron 14 (c.1875+661_1875+851). Similar results were found for the index patient.

As discussed in Section 2.3.3, the minigene assay for the *CACNA1F* variant (NM_005183.4:c.2239+5C>G) revealed a complex collection set of transcripts, with the main variant-induced aberrant transcript being characterized by exon 16 skipping and an elongated exon 17 (Figure 2 and Table 6). Similarly, PCR amplification of exons 15–17 from blood cDNA of the index patient confirmed the main variant-induced transcript (40.8% of the reads) to be characterized by exon 16 skipping and the elongated exon 17 (corresponding to T1 in the minigene results). Additionally, a transcript was detected differing from the WT only by the elongation of exon 17 (T2, 35.7% of the reads), which was not identified in the minigene assay. Finally, a transcript was found with exon 16 skipping, intron 15 retention (c.2118+1_2129), and the elongated exon 17 (19.6%), corresponding to T10 in the minigene assay (Figure A6 and Table A6).

## 3. Discussion

We applied Nanopore deep sequencing for the identification and quantification of aberrant splicing events in minigene and blood-derived cDNA assays due to a diverse set of candidate splicing variants in IRD genes. The method allowed for the characterization of complete transcripts and the identification of rare splicing events. Thirteen out of nineteen variants were found to lead to highly variable levels of aberrant splicing. Five variants led to an alternative splice site used in the main transcript: *ATF6*:c.1096-15G>A, *ATF6*:c.1534-9A>G, *PDE6C*:c.864+1G>A, *POC1B*:c.677-2A>G, and *RPGR*:c.1415-9A>G. Three variants caused exon skipping: *CHM*:c.1413G>C, *PROM1*:c.2358C>T, and *PROM1*:c.2490-2A>G. Multiple aberrant splicing events, including exon skipping and alternative splice sites, were recognized for four variants: *CACNA1F*:c.2239+5C>G, *IMPG2*:c.3423-7_3423-4del, *REEP6*:c.517G>A, and *KIF11*:c.1875+2T>A. Finally, pseudoexon inclusion was found to be the main variant-induced event for a deep-intronic variant in *POC1B* (NM_172240.2:c.1033-327T>A). Conversely, the *ABCA4*:c.573C>T, *ABCA4*:c.5586T>A, *FZD4*:c.313A>G, *TIMP3*:c.205-3117T>C, and *USH2A*:c.652-22287T>C variants had no measurable effect on splicing in these assays. The assay outcomes were used to re-classify variants according to ACMG guidelines (functional evidence PS3/BS3 criterion). This led to a higher class for nine variants, a lower class for four variants, and an unchanged class for six variants (Table 7).

The splicing prediction algorithms included in this study proved to be reliable tools in predicting the nature of variant-induced aberrant splicing, as well as the magnitude of the effect for most variants. In particular, SpliceAI and Pangolin identified and scored accurately the effects of variants *ATF6*:c.1534-9A>G (average splice loss score of 78.5% and 80.6% WT transcript loss measured), *CHM*:c.1413G>C (average splice loss score of 79.5% and 85.1% WT transcript loss measured), *PDE6C*:c.864+1G>A (average splice loss score of 89.5% and 98.0% WT transcript loss measured), *POC1B*:c.677-2A>G (average splice loss score of 92.5% and 92.3% WT transcript loss measured), and *PROM1*:c.2358C>T (average splice loss score of 30.5% and 39.1% WT transcript loss measured). A notable exception is the predictions for *CACNA1F*:c.2239+5C>G, which suggested a loss of WT transcript in the range of 35%, and the use of a cryptic donor site (located at c.2200) in 16% of the transcripts. Transcript quantification resulted in a complete loss of WT transcript in the variant minigene and blood cDNA assays. Additionally, no transcript using the predicted cryptic donor site was identified. Similarly, predictions for *ABCA4*:c.573C>T and *ABCA4*:c.5586T>A, *FZD4*:c.313A>G, *TIMP3*:c.205-3117T>C, and *USH2A*:c.652-22287T>C were mild, and no aberrant splicing could be detected. While the minigene assay results did not support aberrant splicing for these variants, it may be possible that tissue-specific splicing might result in different quantities of alternatively spliced transcript in the relevant tissue.

An important caveat regarding this study is that minigene assays are simplified models of splicing, often limited to small portions of the gene that is being functionally tested. The same applies to blood-derived cDNA assays unless blood is the disease-relevant tissue. Aberrant splicing events detected using these models may not reflect the actual physiological variant-induced effects in the relevant cell type(s) or tissues. It has been previously shown that splicing variants can have different effects and magnitude depending on the model used [47]. While the exact nature of the effect on transcripts may not be extrapolated from these assays alone, the fact that aberrant splicing events are detected is a strong indication that the variant does affect splicing processes. For this reason, minigene assays remain particularly useful for the characterization of variant-induced aberrant splicing for inaccessible tissues, such as the retina. Combining these assays with Nanopore sequencing allows for unparalleled precision in the identification of transcripts and their relative abundance. Additionally, a limitation of the method is represented by its reliance on PCR amplification, which is prone to biases that could confound the results. Nevertheless, this method will help streamline and improve the analysis of novel candidate splicing variants, particularly for complex splicing patterns with multiple transcripts. Understanding the exact nature of aberrant splicing events could be crucial in the development of personalized therapies (e.g., antisense oligonucleotide-based therapies).

## 4. Materials and Methods

### 4.1. Patient Cohort

Index patients were referred to us for genetic testing from large specialized medical centers in Switzerland. Blood samples were collected from index patients and available family members. Written informed consent was obtained from all patients and family members included in this study. This study was conducted in accordance with the 2013 Declaration of Helsinki. A subset of the patients included in this study have been included in previous studies from our group [39,40,41].

### 4.2. Genetic Testing

Genomic DNA (gDNA) was extracted from whole blood in duplicate with the automated Chemagic MSM I system, according to the manufacturer’s specifications (PerkinElmer Chemagen Technologie GmbH, Baesweiler, Germany). Genetic testing strategies included WES, whole-gene sequencing by long-range PCR, or WGS. WES was performed as previously described [48]. Whole-gene sequencing was performed as previously described [40]. WGS was performed as previously described [41].

### 4.3. Variants Annotation and Filtering

Variant Call Format (VCF) files were annotated with the Nirvana (https://illumina.github.io/NirvanaDocumentation/, accessed on 6 October 2021) annotator. The Nirvana output JSON file was converted to a tabular format and filtered with an IRD-associated loci list (Appendix A), as previously described [41]. ACMG classification was performed automatically on Varsome (https://varsome.com/, accessed on 1 July 2024) and Franklin platforms (https://franklin.genoox.com/clinical-db/home, accessed on 1 July 2024).

### 4.4. Splicing Predictions

Several bioinformatic tools were used to predict the effect of candidate variants on splicing. SpliceAI [19], Pangolin [20], SpliceSiteFinder-like (SSF) [21,22], MaxEntScan [23], NNSPLICE [24], and GeneSplicer [25] were used to predict effects on splice sites and branch points. ESEFinder [49], RESCUE-ESE [50], and EX-SKIP [26] were used to examine the effects on ESE and ESS binding sites. To access SpliceAI and Pangolin predictions, the SpliceAI lookup website was used (https://spliceailookup.broadinstitute.org, accessed on 14 June 2024). The splicing prediction module of Alamut^®^ Visual Plus v.1.6.1 (Sophia Genetics, Rolle, Switzerland) was employed to assess the SSF, MaxEntScan, NNSPLICE, GeneSplicer, ESEFinder, and RESCUE-ESE scores. The EX-SKIP website (https://ex-skip.img.cas.cz, accessed on 14 June 2024) was utilized to compare ESE/ESS profiles of reference and variant allele sequences. The SSF, MaxEntScan, NNSPLICE, and GeneSplicer scores were transformed into percentages, and an average “Effect score” was calculated.

Rare variants that may affect splicing identified in patients affected by IRDs were selected for functional assays.

### 4.5. Minigene Assays

The effect of most variants in this study was functionally tested in a cellular system using minigene constructs. Most minigene constructs are based on the previously published pcDNA3.1 backbone (Invitrogen, Carlsbad, CA, USA), containing the genomic region encompassing exons 3 to 5 of the gene *RHO* with an artificial start codon introduced in *RHO* exon 3 [39,48,51,52]. To introduce the genomic region of interest, exon 4 of *RHO* and part of the flanking introns were excised from the construct by digestion, using the restriction enzymes *Pfl*MI and *Eco*NI. The genomic regions of interest were amplified by PCR from patient’s gDNA with Phusion High-Fidelity DNA Polymerase (New England Biolabs, Ipswich, MA, USA) for a total volume of 50 μL, containing 1× Phusion High-Fidelity Buffer, 0.5 μM of each primer, 0.2 mM dNTPs, 0.02 U/μL Phusion High-Fidelity DNA Polymerase, and 10 ng of gDNA. PCR reactions were performed on a Veriti thermal cycler (Applied Biosystems, Foster City, CA, USA) according to the following conditions: 98 °C for 30 s, 35 cycles of 98 °C for 10 s, 58–62 °C (depending on the primers) for 30 s, 72 °C for 5 min, and 72 °C for 10 min. PCR products were verified by electrophoresis on 1% agarose gels.

The minigene constructs for the variants in *ATF6*, *FDZ4*, and *NRL* are not based on the *RHO* backbone. Instead, exons 1 and 2 of the native gene were included. Specifically, for the *ATF6* minigene constructs (one variant located in intron 8 and the other located in intron 12), exons 1 and 2 of *ATF6* and the exon downstream of the variant (exons 9 and 13) were cloned into the pcDNA3.1 backbone (Invitrogen, Carlsbad, CA, USA) using the Takara In-Fusion HD cloning kit (Takara, Kusatsu, Japan). Similarly, the entire coding sequence and UTRs of the *FZD4* gene and *NRL* (NM_006177.3) were inserted into the pcDNA3.1 backbone (Invitrogen, Carlsbad, CA, USA), using the Takara In-Fusion HD cloning kit (Takara Bio, Kusatsu, Japan). The genomic region of interest was amplified by PCR, as described in the previous paragraph.

Sanger and/or long-range PCR sequencing were performed to verify the genotype of the region of interest in selected clones, as previously described [40,48].

The plasmids were transfected into HEK293T cells by Xfect Transfection Reagent (Takara, Kusatsu, Japan), according to the manufacturer’s instructions. Cells were harvested after 24 h and total RNA was isolated, and reverse transcribed into cDNA with the NucleoSpin RNA Plus (Macherey-Nagel, Düren, Germany) and SuperScript III First-Strand Synthesis SuperMix (Invitrogen, Waltham, MA, USA) kits, according to the manufacturer’s instructions.

Primer sequences used for the amplification of the genomic regions of interest are listed in Appendix A.

### 4.6. Blood RNA Assays

Whole blood was collected in PAXgene Blood RNA Tubes (PreAnalytiX, Hombrechtikon, Switzerland) from index patients and family members, when available. The PAXgene Blood RNA Kit (PreAnalytiX, Hombrechtikon, Switzerland) was used to extract total RNA, as previously described [48]. Total RNA was then reverse transcribed into cDNA with the SuperScript III First-Strand Synthesis SuperMix (Invitrogen, Waltham, MA, USA) with oligo(dT)20 primers, according to the manufacturer’s instructions.

### 4.7. Nanopore Sequencing

In the case of *RHO*-backbone constructs, primers binding to *RHO* exons 3 and 5 were used to amplify the minigene-derived transcripts. Primers binding to the T7 promoter region and the *BGH* terminator were used to amplify *ATF6*, *FZD4*, and *NRL* minigene-derived transcripts. These primers also contained adapter sequences for the Nanopore PCR Barcoding Kit SQK-PBK004 (TTTCTGTTGGTGCTGATATTGC-forward primer sequence, and ACTTGCCTGTCGCTCTATCTTC-reverse primer sequence; Oxford Nanopore Technologies, Oxford, UK). Primer sequences are available in Appendix A.

Similarly, primers containing the adapter sequences for the Nanopore PCR Barcoding Kit were designed to amplify the *CACNA1F* exons 15–17 and *KIF11* exons 13–16 regions from whole blood cDNA. Primer sequences are available in Appendix A.

The transcripts of interest were first amplified by PCR in 50 µL volume, according to the Phusion High-Fidelity DNA Polymerase protocol (New England Biolabs, Ipswich, MA, USA), using the GC-Buffer and 100 ng of cDNA with the following conditions: 98 °C for 30 s, 35 cycles of 98 °C for 10 s, 63 °C with the *RHO* primers or 53 °C with T7/*BGH* primers for 30 s, 72 °C for 9 min, and 72 °C for 10 min. PCR products were verified by electrophoresis on 1% agarose gels. PCR reactions were purified with AMPure XP beads (Beckman Coulter Life Sciences, Indianapolis, IN, USA) with a 1:1.5 (PCR: beads) ratio and eluted in 50 µL of 1× Tris-EDTA (TE) buffer (Integrated DNA Technologies, Coralville, IA, USA), according to the manufacturer’s instructions. Concentrations of purified PCRs were measured with the QuBit dsDNA High Sensitivity Assay Kit (Thermofisher Scientific, Waltham, MA, USA). These data were used to dilute each purified PCR in ddH_2_O to 10 ng/µL, for a final volume of 24 µL.

Subsequently, an indexing PCR was performed by adding 25 µL of Long Amp Taq 2X Master Mix (New England Biolabs, Ipswich, MA, USA) and 1 µL of barcoded universal primers with rapid attachment chemistry from the Nanopore PCR Barcoding Kit SQK-PBK004 (Oxford Nanopore Technologies, Oxford, UK), with the following conditions: 94 °C for 1 min, 30 cycles of 94 °C for 30 s, 62 °C for 30 s, 65 °C for 2 min, and 65 °C for 5 min. Indexing PCRs were purified using AMPure XP beads with a 1:1 (PCR: beads) ratio and eluted in 22 µL of Resuspension Buffer (Illumina, San Diego, CA, USA). Concentrations of indexing PCRs were quantified with the QuBit dsDNA High Sensitivity Assay Kit (Thermofisher Scientific, Waltham, MA, USA). The size distribution of PCR products was measured with a Bioanalyzer High-Sensitivity DNA kit on a Bioanalyzer 2100 instrument (Agilent Technologies, Santa Clara, CA, USA). Concentration and size distribution data were used to pool purified PCRs to a total of 50–90 fmol in a final volume of 10 µL. Finally, the rapid 1D sequencing adapters were attached by the addition of 1 µL of RAP to the PCRs pool, which was then incubated for 5 min at room temperature.

The finalized libraries were sequenced with an R9.4.1 (FLO-MIN106D) Flow Cell on a MinION Mk1C instrument (Oxford Nanopore Technologies, Oxford, UK) using the MinKNOW v.23.07.5 software, according to the manufacturer’s instructions.

### 4.8. Nanopore Sequencing Data Analysis

Basecalling was performed to convert pod5 files to FASTQ files, using the wf-basecalling v.1.0.1 workflow on the EPI2ME v.5.1.3 platform (Oxford Nanopore Technologies, Oxford, UK). The resulting FASTQ files were demultiplexed with the Barcoding Analysis module on MinKNOW v.23.07.5 software. Alignment was carried out with minimap2 v2.26, with the “splice” option active [53]. Finally, the alignment file was sorted, indexed, and converted to the BAM format with samtools v.1.18 [54].

### 4.9. Splice Junctions Characterization and Usage Quantification

The JWR_checker.py script from NanoSplicer v1.0 [55] was used to detect splice junctions from the minimap2 alignment results for the minigene assays. The resulting output file was used to identify and quantify high-quality transcripts (reads) from the sequencing data. Briefly, only transcripts characterized by at least one high-quality splice junction (JAQ = 1) were kept. In the case of *RHO*-backbone constructs, only transcripts including both *RHO* exons were considered further. For each transcript identified, the number of reads representing them, and the mean junction quality (JAQ), were calculated. Only transcripts represented by at least 0.5% of the high-quality reads were kept. A construct-specific gff3 file was used to annotate known junctions and exons included in the transcripts and to calculate their length in base pairs. The in-house scripts used to transform the JWR_checker.py outputs are available on GitHub (https://github.com/jordimaggi/Minigene_transcripts_quantification_Nanopore; accessed on 21 July 2024).

When unknown splice sites were detected, the resulting transcript table was manually curated; the location of unknown acceptor and donor sites was verified on Alamut Visual for the existence of cryptic splice sites. If the predictions software on Alamut showed no scores at the splice site location identified during sequencing, the splice junction was assumed to be wrongly called and manually corrected to the most likely nearby splice junction. To visualize the identified transcripts, a gff3 file was created.

## Figures and Tables

**Figure 1 ijms-25-09569-f001:**
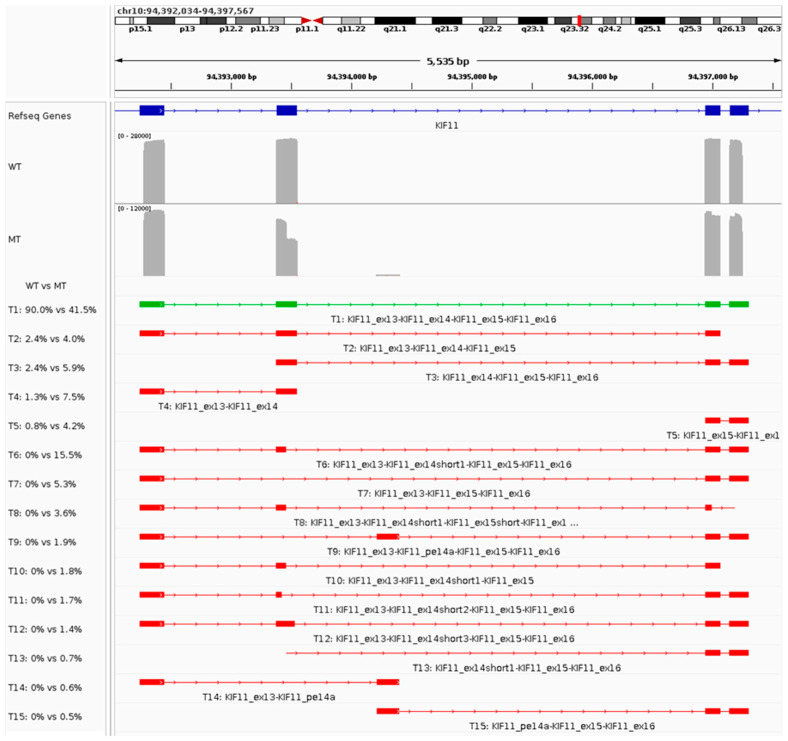
Functional characterization of the *KIF11* variant NM_004523.3:c.1875+2T>A using patient-derived blood cDNA. The panel shows an IGV screenshot highlighting the construct’s characteristics, followed by the coverage plots for the reference (WT) and variant (MT) sequences. An overview of each transcript (name T#) identified and its relative abundance in WT and MT can be seen underneath the coverage plots. The green transcript represents the expected reference (WT) transcript.

**Figure 2 ijms-25-09569-f002:**
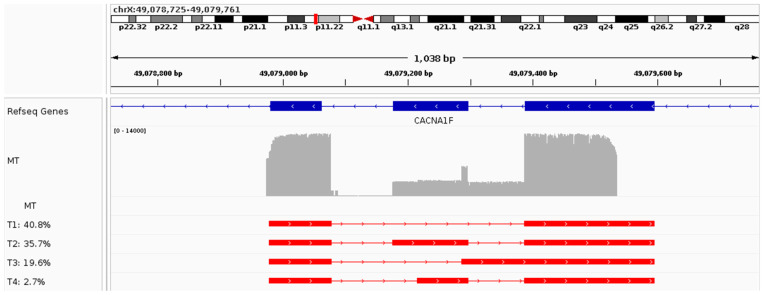
Functional characterization of the *CACNA1F* variant NM_005183.4:c.2239+5C>G using patient-derived blood cDNA. The top panel shows an IGV screenshot highlighting the construct’s characteristics, followed by the coverage plots for variant (MT) sequence. An overview of each transcript (name T#) identified and its relative abundance in MT can be seen underneath the coverage plots.

**Table 2 ijms-25-09569-t002:** In silico splicing predictions summary. The EX-skip prediction column reports the chance (likelihood) of exon skipping computed by EX-skip; it compares the exonic splicing enhancer (ESE) and silencer (ESS) sequences in the reference and variant exons. The last column lists the conclusions based on integrating the predictions for the natural (canonical) splice sites and cryptic splice sites that may be affected by the variants, along with the ESE/ESS profiles.

Gene	Variant (cNomen)	Natural SS	Cryptic SS	EX-Skip Prediction	Conclusion
*ABCA4*	NM_000350.2:c.573C>T	Weakens AS	Unaffected	MT higher chance	Partial exon skipping
*ABCA4*	NM_000350.2:c.5586T>A	Weakens AS	Unaffected	Comparable	Partial exon skipping
*ATF6*	NM_007348.3:c.1096-15G>A	Weakens AS	Creates AS	MT higher chance	Partial usage of alternative AS/Partial exon skipping
*ATF6*	NM_007348.3:c.1534-9A>G	Weakens AS	Creates AS	MT higher chance	Partial usage of alternative AS/Partial exon skipping
*CACNA1F*	NM_005183.4:c.2239+5C>G	Strengthens DS	Unaffected	Comparable	Partial exon skipping
*CHM*	NM_000390.4:c.1413G>C	Strengthens AS and weakens DS	Unaffected	MT higher chance	Partial exon skipping
*FZD4*	NM_012193.4:c.313A>G	Strengthens AS	Creates AS	Comparable	Partial usage of alternative AS
*IMPG2*	NM_016247.4:c.3423-7_3423-4del	Weakens AS	Strengthens AS	WT higher chance	Partial usage of alternative AS/Partial exon skipping
*KIF11*	NM_004523.3:c.1875+2T>A	Weakens DS	Strengthens AS	MT higher chance	Partial usage of alternative DS/Partial exon skipping
*OCA2*	NM_000275.3:c.574-53C>G	Weakens AS	Creates AS	Comparable	Partial exon skipping
*PDE6C*	NM_006204.3:c.864+1G>A	Abolishes DS	Unaffected	Comparable	Partial exon skipping
*POC1B*	NM_172240.2:c.677-2A>G	Abolishes AS	Creates AS	WT higher chance	Partial usage of alternative AS/Partial exon skipping
*POC1B*	NM_172240.2:c.1033-327T>A	Abolishes AS	Creates AS	WT higher chance	Usage of alternative AS
*PROM1*	NM_006017.3:c.2358C>T	Weakens AS and DS	Strengthens AS	MT higher chance	Partial exon skipping
*PROM1*	NM_006017.3:c.2490-2A>G	Abolishes AS	Unaffected	MT higher chance	Partial exon skipping
*REEP6*	NM_001329556.3:c.517G>A	Weakens DS	Weakens AS and strengthens DS	Comparable	Partial usage of alternative DS/Partial exon skipping
*RPGR*	NM_001034853.1:c.1415-9A>G	Weakens AS	Creates AS	MT higher chance	Partial usage of alternative AS
*TIMP3*	NM_000362.4:c.205-3117T>C	Unaffected	Weakens AS	WT higher chance	Partial PE inclusion
*USH2A*	NM_206933.2:c.652-22287T>C	Unaffected	Strengthens AS	WT higher chance	Partial PE inclusion

Abbreviations: cNomen, Human Genome Variation Society (HGVS) cDNA-level nucleotide change nomenclature; SS, splice site; AS, acceptor splice site; DS, donor splice site; and MT, mutant (or variant).

**Table 3 ijms-25-09569-t003:** Minigene constructs.

Minigene Construct	Variant (cNomen)	Insert Genomic Coordinates (hg19)	Expected Major (WT) Transcript Length (bp)
RHO_minigene_ABCA4_int4-6	NM_000350.2:c.573C>T	chr1:94563849-94569762	445
RHO_minigene_ABCA4_int38-41	NM_000350.2:c.5586T>A	chr1:94473946-94478346	494
RHO_minigene_ATF6_int8-9	NM_007348.3:c.1096-15G>A	chr1:161790639-161791116	211
ATF6_minigene_ex1-2-9	NM_007348.3:c.1096-15G>A	chr1:161735948-161736532 chr1:161747734-161748410 chr1:161790560-161791251	850
ATF6_minigene_ex1-2-13	NM_007348.3:c.1534-9A>G	chr1:161735948-161736532 chr1:161747734-161748410 chr1:161829697-161830367	795
RHO_minigene_CACNA1F_int14-18	NM_005183.4:c.2239+5C>G	chrX:49077239-49079781	576
RHO_minigene_CHM_int9-11	NM_000390.4:c.1413G>C	chrX:85155080-85156836	288
FZD4_minigene_ex1-2	NM_012193.4:c.313A>G	chr11:86661826-86666316	2247
RHO_minigene_IMPG2_int15-18	NM_016247.4:c.3423-7_3423-4del	chr3:100947361-100950310	599
RHO_minigene_IMPG2_int16-17	NM_016247.4:c.3423-7_3423-4del	chr3:100947853-100949413	330
RHO_minigene_OCA2_int5-7	NM_000275.3:c.574-53C>G	chr15:28263150-28268457	353
RHO_minigene_PDE6C_int3-4	NM_006204.3:c.864+1G>A	chr10:95381106-95382819	260
RHO_minigene_POC1B_int6-7	NM_172240.2:c.677-2A>G	chr12:89863350-89865358	253
RHO_minigene_POC1B_int9-10	NM_172240.2:c.1033-327T>A	chr12:89852809-89855060	200
RHO_minigene_PROM1_int20-23	NM_006017.3:c.2358C>T	chr4:15983331-15988085	362
RHO_minigene_PROM1_int23-26	NM_006017.3:c.2490-2A>G	chr4:15980451-15982543	328
RHO_minigene_REEP6_int1-5	NM_001329556.3:c.517G>A	chr19:1494671-1496679	521
RHO_minigene_RPGR_int10-13	NM_001034853.1:c.1415-9A>G	chrX:38149931-38157030	446
RHO_minigene_TIMP3_int1-3	NM_000362.4:c.205-3117T>C	chr22:33244858-33253580	314
RHO_minigene_USH2A_int3	NM_206933.2:c.652-22287T>C	chr1:216558743-216563578	119

Abbreviations: cNomen, Human Genome Variation Society (HGVS) cDNA-level nucleotide change nomenclature.

**Table 4 ijms-25-09569-t004:** Minigene assay results summary. The aberrant splicing events column reports the main aberrant splicing events induced or favored by the variant. The last column lists the difference in relative abundance of the expected reference (WT) transcript between variant and reference minigenes.

Minigene Construct	Variant (cNomen)	Aberrant Splicing Events	Δ WT Transcript (%)
RHO_minigene_ABCA4_int4-6	NM_000350.2:c.573C>T	NA	+5.3
RHO_minigene_ABCA4_int38-41	NM_000350.2:c.5586T>A	NA	−0.7
RHO_minigene_ATF6_int8-9	NM_007348.3:c.1096-15G>A	Alternative AS	+1.4
ATF6_minigene_ex1-2-9	NM_007348.3:c.1096-15G>A	Alternative AS	−25.2
ATF6_minigene_ex1-2-13	NM_007348.3:c.1534-9A>G	Alternative AS	−80.6
RHO_minigene_CACNA1F_int14-18	NM_005183.4:c.2239+5C>G	Exon skipping/ alternative AS	−11.2
RHO_minigene_CHM_int9-11	NM_000390.4:c.1413G>C	Exon skipping	−85.1
FZD4_minigene_ex1-2	NM_012193.4:c.313A>G	NA	+1.8
RHO_minigene_IMPG2_int15-18	NM_016247.4:c.3423-7_3423-4del	Exon skipping	−1.1
RHO_minigene_IMPG2_int16-17	NM_016247.4:c.3423-7_3423-4del	Exon skipping/ alternative AS	−47.0
RHO_minigene_OCA2_int5-7	NM_000275.3:c.574-53C>G	NA	+7.3
RHO_minigene_PDE6C_int3-4	NM_006204.3:c.864+1G>A	Alternative DS	−98.0
RHO_minigene_POC1B_int6-7	NM_172240.2:c.677-2A>G	Alternative AS	−92.3
RHO_minigene_POC1B_int9-10	NM_172240.2:c.1033-327T>A	PE inclusion	−48.0
RHO_minigene_PROM1_int20-23	NM_006017.3:c.2358C>T	Exon skipping	−39.1
RHO_minigene_PROM1_int23-26	NM_006017.3:c.2490-2A>G	Exon skipping	−3.7
RHO_minigene_REEP6_int1-5	NM_001329556.3:c.517G>A	Exon skipping/ alternative DS	−55.9
RHO_minigene_RPGR_int10-13	NM_001034853.1:c.1415-9A>G	Alternative AS	−55.2
RHO_minigene_TIMP3_int1-3	NM_000362.4:c.205-3117T>C	NA	0.0
RHO_minigene_USH2A_int3	NM_206933.2:c.652-22287T>C	NA	−0.6

Abbreviations: cNomen, Human Genome Variation Society (HGVS) cDNA-level nucleotide change nomenclature; Δ, delta; AS, acceptor splice site; DS, donor splice site; PE, pseudoexon; and NA, not applicable.

**Table 5 ijms-25-09569-t005:** Transcript identification and quantification for the *KIF11* variant M_004523.3:c.1875+2T>A for reference (WT) and variant (MT) sequences. The table lists the transcripts identified, along with their characteristics, such as length, their relative abundance in reference (WT) and variant (MT) minigenes, the difference (delta) in relative abundance between MT and WT sequencing results, the absolute number of reads representing each transcript, and the effect on the transcript. The table is sorted by relative abundance.

	Transcript	Length	WT (%)	MT (%)	Δ MT-WT (%)	Counts WT	Counts MT	Effect on Transcript
T1	KIF11_ex13-KIF11_ex14-KIF11_ex15-KIF11_ex16	666 bp	89.98	41.53	−48.45	16,855	3205	WT
T2	KIF11_ex13-KIF11_ex14-KIF11_ex15	507 bp	2.38	3.99	1.61	446	308	ex16 skip
T3	KIF11_ex14-KIF11_ex15-KIF11_ex16	458 bp	2.37	5.88	3.51	444	454	ex13 skip
T4	KIF11_ex13-KIF11_ex14	381 bp	1.28	7.51	6.23	239	580	ex15-16 skip
T5	KIF11_ex15-KIF11_ex16	285 bp	0.81	4.17	3.36	152	322	ex13-14 skip
T6	KIF11_ex13-KIF11_ex14short1-KIF11_ex15-KIF11_ex16	576 bp	0	15.5	15.5	0	1197	altDS_ex14short
T7	KIF11_ex13-KIF11_ex15-KIF11_ex16	493 bp	0	5.33	5.33	0	412	ex14 skip
T8	KIF11_ex13-KIF11_ex14short1-KIF11_ex15short-KIF11_ex16short	466 bp	0	3.55	3.55	0	274	altDS_ex14short + altDS_ex15short + altAS_ex16short
T9	KIF11_ex13-KIF11_pe14a-KIF11_ex15-KIF11_ex16	683 bp	0	1.88	1.88	0	145	ex14 skip + pe14a
T10	KIF11_ex13-KIF11_ex14short1-KIF11_ex15	417 bp	0	1.83	1.83	0	141	altDS_ex14short + ex16 skip
T11	KIF11_ex13-KIF11_ex14short2-KIF11_ex15-KIF11_ex16	540 bp	0	1.66	1.66	0	128	altDS_ex14short
T12	KIF11_ex13-KIF11_ex14short3-KIF11_ex15-KIF11_ex16	651 bp	0	1.39	1.39	0	107	altDS_ex14short
T13	KIF11_ex14short1-KIF11_ex15-KIF11_ex16	368 bp	0	0.69	0.69	0	53	ex13 skip + altDS_ex14short
T14	KIF11_ex13-KIF11_pe14a	398 bp	0	0.58	0.58	0	45	pe14a + ex14-16 skip
T15	KIF11_pe14a-KIF11_ex15-KIF11_ex16	475 bp	0	0.53	0.53	0	41	ex13-14 skip + pe14a

Abbreviations: WT, wildtype (or reference); MT, mutant (or variant); ex, exon; alt, alternative; AS, acceptor splice site; DS, donor splice site; pe, pseudoexon; bp, base pairs; Δ, delta.

**Table 6 ijms-25-09569-t006:** Transcript identification and quantification for the *CANCA1F* variant M_005183.4:c.2239+5C>G for the variant (MT) sequence. The table lists the transcripts identified, along with their characteristics, such as length, their relative abundance in the variant (MT) sequence, the absolute number of reads representing each transcript, and the effect on the transcript. The table is sorted by relative abundance.

	Transcript	Length	MT (%)	Counts MT	Effect on Transcript
T1	CACNA1F_ex15-CACNA1F_ex17long	307 bp	40.76	1861	ex16 skip + altAS_ex17long
T2	CACNA1F_ex15-CACNA1F_ex16-CACNA1F_ex17long	428 bp	35.7	1630	altAS_ex17long
T3	CACNA1F_int16-CACNA1F_ex17long	407 bp	19.56	893	int16 retention + altAS_ex17long
T4	CACNA1F_ex15-CACNA1F_ex16short-CACNA1F_ex17long	388 bp	2.72	124	altDS_ex16short + altAS_ex17long

Abbreviations: MT, mutant (or variant); ex, exon; int, intron; alt, alternative; AS, acceptor splice site; DS, donor splice site; bp, base pairs.

**Table 7 ijms-25-09569-t007:** Study outcome overview with adjusted ACMG classification. * ACMG recomputed on the Franklin platform by manually curating Functional Studies evidence based on this study results (Evidence categories PS3/BS3). Functional Studies evidence set on “Strong” for most assays. ^1^ Functional Studies evidence set on “Moderate” based on aberrant splicing evidence from assays. ^2^ Functional Studies evidence set on “Very strong” based on aberrant splicing evidence from assays. ^3^ ACMG classification is unchanged because the variant affects the protein function by altering the amino acid sequence.

Gene	Variant	Franklin ACMG Class	Splicing Predictions	Splicing Assay	New ACMG Class *
*ABCA4*	NM_000350.2:c.573C>T	3	Partial exon skipping	No effect	2
*ABCA4*	NM_000350.2:c.5586T>A	2	Partial exon skipping	No effect	2
*ATF6*	NM_007348.3:c.1096-15G>A	3	Partial usage of alternative AS/Partial exon skipping	Partial usage of alternative AS	3 ^1^
*ATF6*	NM_007348.3:c.1534-9A>G	3	Partial usage of alternative AS/Partial exon skipping	Usage of alternative AS	4
*CACNA1F*	NM_005183.4:c.2239+5C>G	3	Partial exon skipping	Multiple effects	4
*CHM*	NM_000390.4:c.1413G>C	3	Partial exon skipping	Exon skipping	5 ^2^
*FZD4*	NM_012193.4:c.313A>G	5	Partial usage of alternative AS	No effect	5 ^3^
*IMPG2*	NM_016247.4:c.3423-7_3423-4del	4	Partial usage of alternative AS/Partial exon skipping	Multiple effects	5
*KIF11*	NM_004523.3:c.1875+2T>A	4	Partial usage of alternative DS/Partial exon skipping	Multiple effects	5 ^1^
*OCA2*	NM_000275.3:c.574-53C>G	2	Partial exon skipping	No effect	1
*PDE6C*	NM_006204.3:c.864+1G>A	5	Partial exon skipping	Usage of alternative DS	5 ^2^
*POC1B*	NM_172240.2:c.677-2A>G	4	Partial usage of alternative AS/Partial exon skipping	Usage of alternative AS	5
*POC1B*	NM_172240.2:c.1033-327T>A	3	Usage of alternative AS	Usage of alternative AS	4
*PROM1*	NM_006017.3:c.2358C>T	1	Partial exon skipping	Partial exon skipping	1
*PROM1*	NM_006017.3:c.2490-2A>G	5	Partial exon skipping	Partial exon skipping	5 ^1^
*REEP6*	NM_001329556.3:c.517G>A	3	Partial usage of alternative DS/Partial exon skipping	Multiple effects	4
*RPGR*	NM_001034853.1:c.1415-9A>G	4	Partial usage of alternative AS	Usage of alternative AS	5
*TIMP3*	NM_000362.4:c.205-3117T>C	3	Partial PE inclusion	No effect	2
*USH2A*	NM_206933.2:c.652-22287T>C	3	Partial PE inclusion	No effect	2

Abbreviations: ACMG, American College of Medical Genetics and Genomics guidelines; VUS, variant of unknown significance; AS, acceptor splice site; DS, donor splice site; and PE, pseudoexon.

## Data Availability

The original raw data (FASTQ files) used in the study are openly available in Zenodo at 10.5281/zenodo.13143657.

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
