# Peer review of "Nanopore Deep Sequencing as a Tool to Characterize and Quantify Aberrant Splicing Caused by Variants in Inherited Retinal Dystrophy Genes"

_ijms, 2024, doi:10.3390/ijms25179569_

Round 1

Reviewer 1 Report

Comments and Suggestions for Authors

The manuscript by Maggi et al. presents a study using nanopore deep sequencing to characterize and quantify aberrant splicing events caused by variants in genes associated with inherited retinal dystrophies. This study demonstrates that nanopore sequencing enabled the identification and precise quantification of full-length transcripts, aligning with in silico predictions. Additionally, the method reliably detected low-abundant transcripts that conventional strategies, such as RT-PCR followed by Sanger sequencing, would not be able to identify.

The manuscript is well-organized and conclusive. Specific comments/suggestions to further improve the manuscript prior to acceptance are as follows:

1.     Please include a table that details the demographics of the patients would further enhance the study.

2.     Can the authors outline the limitations of this study and the next steps for research based on the study’s findings?

Reviewer 2 Report

Comments and Suggestions for Authors

Jordi Maggi et al. presented their study on splicing events caused by genetic variants in IRD (Inherited Retinal Dystrophies) in humans. The authors first predicted the impact of the selected variants on splicing and then validated these alternative splicing events using Nanopore deep sequencing in combination with a minigene assay and peripheral blood cDNA. After reviewing the paper, my suggestions are:

1.        The author showed 19 splicing events in 15 IRD-associated genes, but validated these genes in vitro and blood samples, are these genes expressed in human Rod and RPE?

2.        The author needs to discuss the function of these candidate genes in IRD.

3.        How many replicates for each sequencing sample? The author need to show the results by each replicate and the statistic significance in all the figures.

4.        I recommand that the authors show the Sashimi plots for these alternative splicing events instead of the IGV tracks in all the figures.
